# Mountains as a Global Heritage: Arguments for Conserving the Natural Diversity of Mountain Regions

**Abhik Chakraborty**

Faculty of Tourism, Wakayama University, 930 Sakaedani, Wakayama 640-8510, Japan;
abhich78@wakayama-u.ac.jp or abhichkro@gmail.com

**Abstract:** This concise review posits the urgent need for conserving the natural diversity of mountain environments by envisioning mountains as a global natural heritage. Mountains are recognized as cradles of biodiversity and for their important ecosystem services. Mountains also constitute the second most popular outdoor destination category at the global level after islands and beaches. However, in the current age of accelerating global environmental change, mountain systems face unprecedented change in their ecological characteristics, and consequent effects will extend to the millions who depend directly on ecosystem services from mountains. Moreover, growing tourism is putting fragile mountain ecosystems under increasing stress. This situation requires scientists and mountain area management stakeholders to come together in order to protect mountains as a global heritage. By underlining the salient natural diversity characteristics of mountains and their relevance for understanding global environmental change, this critical review argues that it is important to appreciate both biotic and abiotic diversity features of mountains in order to create a notion of mountains as a shared heritage for humanity. Accordingly, the development of soft infrastructure that can communicate the essence of mountain destinations and a committed network of scientists and tourism scholars working together at the global level are required for safeguarding this shared heritage.

**Keywords:** mountains; natural diversity; abiotic and biotic elements; anthropogenic change; global heritage

## 1. Introduction

This concise and selective review posits the urgent need for conceptualizing mountains as a shared planetary heritage worthy of conservation by drawing on recent scientific literature on threats faced by mountains across the world. It emphasizes the value of natural diversity (encompassing both abiotic and biotic elements) of mountains and highlights the importance of mountains at the global level. It takes a form akin to a "position paper" that advances a specific point (i.e., the pressing need for safeguarding the natural diversity of mountains at a global level), its focus remains limited to that aspect, and it presents a selective overview of literature sources pertaining to that angle. Accordingly, the sections briefly outline the current threat scenario, the broader significance of mountains beyond their immediate physical parameters, and the need for safeguarding mountains as heritage. The review advances the position that as there is no international conservation scheme to protect the biophysical integrity of mountains for its intrinsic value, an alliance of scientists, conservation professionals, tourism planners, and communication experts is urgently needed, which, in collaboration with local societies, can address the ongoing degradation of mountain environments.

## 2. Mountains as Storehouses of Natural Diversity under Threat

Mountains are noted for their important role as cradles of biodiversity [1–3]. It is also well known that mountains are the sources of all major drainage systems of the planet [4,5]. Lately, mountains have been assessed for their rich geodiversity [6]—which is defined as the abiotic diversity of planet earth [7]. The most notable properties of mountains that link abiotic and biotic diversity are their steep gradient and associated climatic variations, which effectively compress "life zones" and create a large number of local niches that support different biota [8]. Moreover, recent case studies have shed light on how geological properties, such as parent rock type and weathering pathways, explain local biodiversity in mountains [9]. In a recent paper, Rahbek et al. (2019) visualized the evolution of mountain environments in the form of a schematic diagram that accounts for two major abiotic drivers—i.e., orogenic processes operating across deep geological time and climatic oscillations of glacials and interglacials spread over the Quaternary Period—that have left lasting imprints on the biotic environments of mountains [10] (Figure 1).

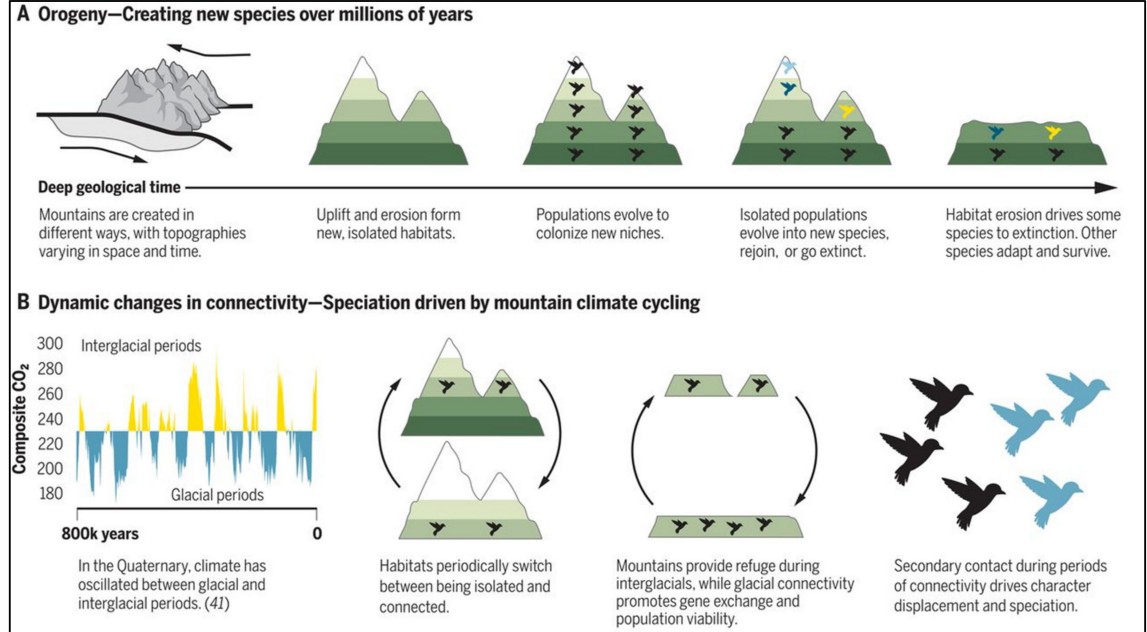

**Figure 1.** Schematic diagram showing two major abiotic factors, orogeny and climatic oscillations, engendering mountain biodiversity over deep time. From Rahbek et al. [10]. Reprinted with permission from AAAS.

Together, the rich biodiversity and geodiversity of mountains make them important storehouses of natural diversity on the planet [9]. Therefore, mountains are not only important for their biodiversity, but are also irreplaceable for their abiotic features. This conceptualization paves the way for recognizing the value of their natural diversity [11]. In this paper, the term "natural diversity" is meant to encompass both abiotic and biotic features of the environment [11,12]. The term "mountain regions" refers to the 12.5% of the planet's terrestrial surface outside Antarctica that is defined as mountain areas [8,13]. The distribution of global mountain regions following Körner et al. (2017) [13] is available in Figure 2.

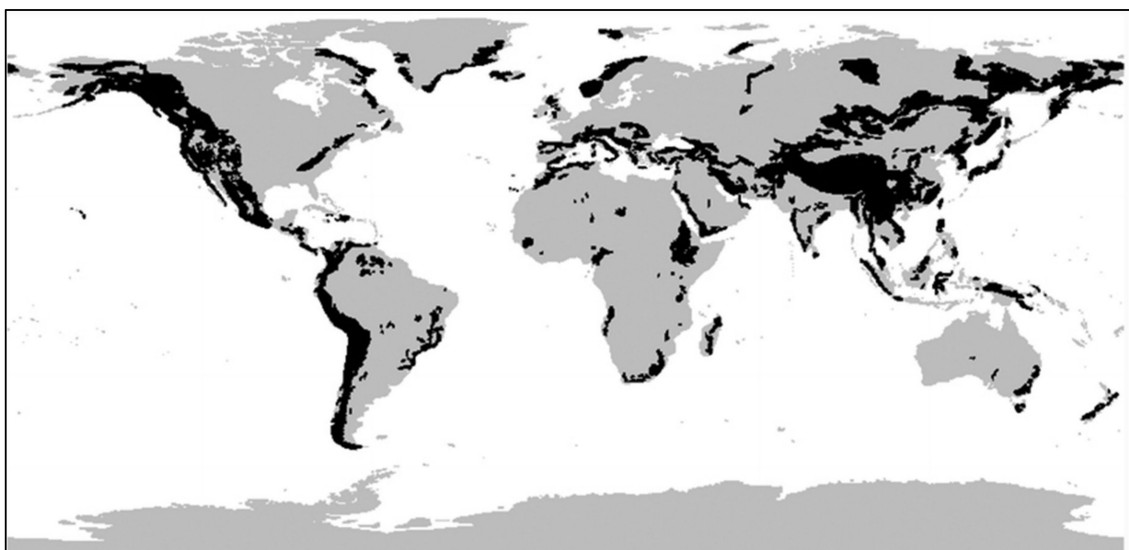

**Figure 2.** Global distribution of mountain systems (areas shaded in black). Figure courtesy of Körner et al. (2017).

However, in the current era of pervasive anthropogenic modifications of important parameters of the earth system, mountains as storehouses of important natural diversity are under incremental threat [14]. There are two broad categories of stressors on mountain systems: direct stressors related to land conversion, expansion of visitation, and appropriation of resources [15–17]; and more diffuse yet powerful stressors such as global climatic change that will disproportionately affect mountains [18–20]. Impacts converge on the loss of glacial ice volume and colonization of formerly cold-adapted niches by lowland species (implying the effective collapse of the densely stacked and finely balanced "life zones") [20,21] and loss of spatial heterogeneity and dynamism of natural processes, notably at the mountain catchment level [22,23]. Consistent loss of ice and snowpack volume is reported from all high mountain systems with the exception of the Karakorum: Wouters et al.'s (2019) global analysis indicated rapid glacial mass loss in high-mountain regions between 2002 and 2016 [24]. This trend is supported by observations of the Andean glaciers between 2000 and 2018 [25], as well as by reports of steady mass loss of Himalayan glaciers since the 1970s [26]. In the Himalayas, a doubling of glacial mass loss was observed during 2000–2016 compared to 1975–2000 [27]. The situation is alarming both for cold-adapted endemic biota and the 1.9 billion people who depend on freshwater from mountains [5]. Milner et al. (2017) detailed the planetary repercussions of glacial melt by analyzing snowmelt-dominated drainage systems and noted possible far-reaching negative effects on provisioning, regulating, and cultural ecosystem services derived from those systems [28]. Reduction of ice volume and ice/snow cover loss in the mountains due to anthropogenic alteration of the climate is problematic because ice/snow cover and glacial volume all constitute important aspects of the abiotic diversity of mountains. Rapid change to these elements forces the system out of the range of mechanisms that evolved in response to natural climatic variations in geological time, with the resultant amplification of extinction risk particularly for cold-adapted biota. In addition, extensive human modification of upstream–downstream connectivity in mountain regions currently constrains the ability of natural systems to recover from shocks [29,30].

It is also worth noting that mountains form the second most popular destination category after islands and coastal areas, and collectively, mountains accommodate 15%–20% of global annual visitations [31]. Yet, mountain destinations remain far short of sustainability: there is widespread poverty and vulnerability of local residents, as well as fragmentation of the natural environment [32,33]. Despite numerous studies voicing concern about these challenges, there is currently no explicit commitment to safeguard the natural environment of mountain systems by adopting tangible goals. In the meantime, expansion of tourism is continually affecting sensitive ecosystems and protected area characteristics [34,35].

## 3. The Broader Significance of Mountains

It is worth stressing the importance of mountains beyond their immediate physical boundaries. Sharma et al. (2019) summed up the net contribution of mountains to biodiversity by noting that mountains house 25% of all terrestrial biodiversity and nearly 50% of all terrestrial biodiversity hotspots and benefit nearly 50% of the human population through various ecosystem services [36]. As Hoorn et al. (2010) demonstrated, the rise of the Andean mountain system contributed to the development of the Amazonian biodiversity hotspot over geological time through the alteration of continent-level drainage and climate patterns [37]. The Andean mountain system itself is a storehouse of endemism—nearly 45,000 plant species with an approximate 45% endemism rate were reported [38], and the imprint of the episodic rise of the Andean system on both Andean and continental vegetation was analyzed [39]. Similarly, the rise of the Himalayan system has modified the South Asian monsoon, resulting in extensive influence on biotic communities across Asia [40,41]. Mountain systems therefore exert a wide influence across riverine and forest biota and play vital supporting roles by supplying nutrients for a range of ecosystems at the continent level [42]. Accordingly, mountains should be appreciated both as vital drivers and refugia of biodiversity, providers of globally important resources, and for their sustenance of rich local cultures. Debarbieux and Price (2008) provided a pertinent pointer regarding the importance of mountains as a global common good and proposed that mountains should be conceptualized as a unique type of common good, i.e., "global common regions", owing to their geographical dimension [43].

The important role of connectivity between mountains and lowland areas deserves both research and practical conservation focus. Recently, Martín-López et al. (2019) provided a synthesis of the evolving research trajectory regarding mountains pertaining to the Nature's Contributions to People (NCP) angle [44]. The authors observed that although research has become increasingly interdisciplinary over the years, the focus on biodiversity, direct drivers of change, and regulating ecosystem services (NCP) has somewhat diminished. Regarding specific mountain systems, spatial connectivity among important ecosystem services was analyzed in a recent study on the European Alps by Schirpke et al. (2019) [45]. The authors pointed out that the predominant pathway of spatial interactions is from high alpine to downstream areas and that the flows of vital ecosystem services such as freshwater provision and outdoor recreation (i.e., ecosystem services with high market value) are global in scale. The study also identified the high popularity of plant and animal species among visitors from a range of European countries. Outdoor recreation is a major aspect of cultural ecosystem services [46], and it can be pointed out that for mountain destinations, primary assets for recreation are biophysical (such as topographic and climatic features). While demand for ecosystem services from mountains is steadily increasing, primarily due to the growth in human population that necessitates further demand for freshwater, living, and amenity space, the ability of mountain systems to provide these services is increasingly constrained as humanity's growing footprint is squeezing heterogeneity and integrity out of mountain landscapes. In addition, vital ecosystem connectivity between mountains and lowland areas is increasingly compromised due to the construction of hydro-dams [22,47]. The conundrum was summed up by Grêt-Regamey et al. (2012) with the observation that the total demand for ecosystem services from mountains should not outstrip the total supply of such services [48]. Under the scenario of shrinking glaciers and collapsing life zones, these primary assets will shrink rapidly with cascading effects on the ecosystem services they generate. Unfortunately, while there is significant interest in observing and curating exotic alpine biota, there is no comparable effort to protect the biophysical integrity of these systems.

For effective conservation of key biophysical processes, it will be necessary to look beyond conserving specific parts of mountains and embrace the totality of natural processes that connect ridgelines to lowland areas. Currently, land use change and construction of hydro-dams form major causes of fragmentation of upland–lowland linkages [49–51]. As mountains are connected with watersheds and lowland environments in time and space, even national park boundaries may not be adequate for safeguarding the integrity of biophysical processes and wildlife movement between

mountains, watersheds, and surrounding landscapes [52]. This situation requires the formulation of conservation plans that can address issues such as the large-scale spatial connectivity of mountains, streams, and lowlands and the integrity of natural space that is key for ecosystem survival [53]. There are some initiatives that attempt to address these issues as outlined in Section 4, but a concerted global effort based on recognition of the vital importance of such connectivity will be required.

## 4. Argument for Mountains as a Global Natural Heritage

There are several notable global initiatives that deal with the natural and cultural environments of mountains. Mountains are the focal area for the activities of the Mountains Specialist Group of the World Commission on Protected Areas of the International Union for Conservation of Nature (IUCN-WCPA) [54]; there is a Mountain Partnership under the UN's Food and Agricultural Organization (FAO) [55]; and mountains are mentioned in the Sustainable Development Goals (SDGs) of the UN as Target 15.4 [56]. In addition, mountains occupy a significant part of the global protected area network [57]. However, the featuring of mountains in global agendas does not guarantee effective conservation of their biophysical aspects, and the formulation of the SDGs reflects a development-oriented vision [58] that considers mountains important for human development [56,59] but not necessarily for their intrinsic value.

This article specifically puts forward the argument that under the ongoing rapid environmental change, mountains should be conceptualized as a global natural heritage shared by all continents and people. This argument extends previous calls for recognizing the global importance of mountains, such as the global common region angle proposed by Debarbieux and Price [43]. Though the natural angle is particularly highlighted here, it is possible to extend this vision to address both natural and cultural angles. An important insight in this regard was provided by Catalan et al. (2017), where they claimed that mountain regions can be conceptualized as suitable places to study nature's response to anthropogenic environmental change [60].

In order to specifically address the conservation of mountain areas as a global natural heritage, careful assessment of their natural diversity characteristics, integrity of biophysical processes that engender that diversity in geological time, and current change pathways will be required. In addition, large contiguous mountain areas that are relatively free from land conversion and development pressures should be safeguarded as areas of high scientific interest as well as places that are intrinsically important for geodiversity, biodiversity, important ecosystem services, and natural beauty. In this regard, the Global Mountain Biodiversity Assessment (GMBA)—while focusing on the biotic environment—provides a platform for international and multidisciplinary collaboration on assessment and conservation of mountain biodiversity and sustainable development of highland regions [61]. The aforementioned IUCN-WCPA (especially the Mountains Specialist Group) is engaged with conservation and sustainable management of mountainous protected areas, and the Connectivity Conservation Group of the IUCN-WCPA deals with landscape-level linkages [62]; together, these IUCN bodies can provide a communication space for conservation scientists, planners, and managers.

With respect to ground-level conservation activities dedicated to safeguarding specific large mountain areas and their ecosystems, the Sky Islands Alliance [63,64] and the Yellowstone to Yukon Conservation Initiative [65,66] can serve as instructive examples. The Sky Islands Alliance is dedicated to conserving wildlife, open spaces, and water resources in the Madrean Archipelago biodiversity hotspot of northwestern Mexico and the southwestern United States, and the Yellowstone to Yukon Conservation Initiative aims to conserve a continent-scale ecosystem spanning the Greater Yellowstone Ecosystem of the US and Canada's Yukon Watershed. Another particularly important vision is embodied in the Canadian Rocky Mountain Parks World Heritage Site (which includes Banff, Jasper, Kootenay, and Yoho National Parks, as well as three regional parks) [67]. Sandford (2010) claimed that the successful registration of the site signaled an enduring cultural achievement and that in setting aside the spine of the Canadian Rockies as a wild space, the vital ecological functions of that area were recognized [67]. Notably, a strong sense of wonder and an attachment to the place were

outlined in this work as essential for maintaining the integrity of that heritage landscape. Taking a cue from this, I argue that an affective association, a sense of cultural belonging, and an appreciation of the diversity and functional integrity of mountain systems are essential for their conservation as a global heritage. It is worth stressing that the dynamic nature of mountain environments—exemplified through vigorous uplift and erosion regimes that occasionally create hazards and inconveniences for humans—is also the most defining characteristic of mountains as a global heritage [14]. This realization is important for appreciating the value of mountain systems on a changing planet. Large conservation initiatives involving mountain systems should be upscaled to incorporate relevant watershed and forest boundaries, and coordination between initiatives from different nations and continents will also be required, because, as Foreman (2013) argued, without upscaling and broader linkages, even large conservation initiatives will remain sporadic and islandic in nature, unable to stem the overall degradation of mountain systems [63]. The need to upscale existing world heritage sites to ensure inter-site connectivity and safeguard wilderness value was described by Kormos et al. (2016) from a species movement angle [68]. This approach can be extended and complemented with Santana's (2019) argument [11] for approaching conservation from the perspective of "natural diversity" rather than focusing solely on species. As mountain systems offer a rich array of geological records, spectacular relief, and climatic and drainage peculiarities, their total value should address these features as well. A possible pathway for extending protection to mountain environments and furthering affective association could be found through the appreciation of large contiguous mountainous sites registered under UNESCO's World Natural Heritage properties for their wilderness aspects and ecological functions. In fact, there are also possibilities to extend such protected areas to incorporate mountain-downstream or mountain-coastal connectivities [68]. Other UNESCO initiatives, such as the Man and Biosphere Reserve [69] and the Ramsar Convention for wetland conservation [70], can also be tapped into for expanding the scope of such conservation. Such undertakings will require the involvement of experts from geological and biological sciences, dedicated professionals with wildlife monitoring skills, and social scientists, tourism researchers, planners, and skilled communication personnel who are able to spread the message out into the wider public. Scientists, area managers, and professionals may liaise with the IUCN's Mountains Specialist Group in order to inform the World Commission on Protected Areas about the priorities for effective conservation of mountain landscapes that encapsulate natural diversity and biophysical processes of exceptional value. Such a network of scientists and area managers dedicated to assessing, maintaining, and communicating both the fine-scale heterogeneity and the broad contours of mountain environments should also be able to promote a new social value of understanding mountains as threatened systems. This value, in turn, can incentivize soft infrastructure development helpful for conservation and education. Soft infrastructure is used here as an umbrella term for technological inputs such as camera or remote sensing for monitoring, communication inputs such as videos and lectures provided in visitor centers for interpretation, and the social and economic investment required to this end. An initiative supported by the WWF and Conservation International to capture wildlife movement through the noninvasive technique of camera trapping has recently garnered international attention [71]. Such soft infrastructure can be adapted to mountain conditions and can be effective for generating data as evidenced by the monitoring of the elusive snow leopard, which is a keystone species in the Himalayan region [72]. Interpretation of data by skilled communicators can bolster knowledge and awareness of symbolic species and biophysical aspects, such as drainage, snow-ice dynamics, and beauty of mountain landscapes.

Finally, it remains to be noted that governance of mountain regions remains key for implementing research and conservation. As analyzed by Price (2015), formal governance structures can take many years to form and are collectives of many different stakeholders, and in order to deliver effective governance of mountain regions, a transnational governance framework is required [73]. Although there are multiple challenges in bringing together different stakeholder groups, specific focus is seen as effective for the coalescing of stakeholder interest [73]. It is envisaged that a theme such as the natural

diversity of mountains and its change pathways can serve as a pertinent point for the convergence of the social capital needed for such transnational governance initiatives.

## 5. Conclusions

This concise review presented an overview of the global level threat scenario affecting mountain systems from the recent scientific literature and emphasized the need to value mountain systems for their endemic diversity as well as their numerous supporting roles. It was observed that while several studies provide important knowledge of the current condition of mountains and adjacent ecosystems, the overall degradation of mountain environments continues, as there is no global conservation scheme to protect the biophysical integrity of mountain systems. Mountains all over the world are currently under incremental threat, due to ongoing fragmentation of their natural diversity resulting from direct and diffuse stressors related to land conversion, intensive resource appropriation, rupture of landscape connectivity, and climate change. Accordingly, the important argument of viewing mountain systems across the planet as a global heritage that merits holistic and ambitious conservation schemes was put forth. Such an association can materialize through an alliance of scientists, conservation professionals, tourism planners, and communication personnel, and it could help to foster a new social value that speaks for the dynamism and heterogeneity of mountain systems that support diversity of life on earth.

**Funding:** This research was funded by the Japan Society for the Promotion of Science, grant number 19K20567.

**Acknowledgments:** The author wishes to thank the reviewers for their constructive feedback, which made this paper better.

**Conflicts of Interest:** The author declares no conflict of interest.

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
