# Peer review of "Mountains as a Global Heritage: Arguments for Conserving the Natural Diversity of Mountain Regions"

_heritage, doi:10.3390/heritage3020012_

Round 1

Reviewer 1 Report

- In the abstract it is necessary to show the goal (s) of the work.

- The paper presents interesting ideas about changes in the mountains that affect all the inhabitants of the Earth, so the topic of the paper is topical.
- The term "Mountain Regions" appears in the title but the text does not explain exactly what it means.
- I don't see why reducing the ice cover would be a threat to the mountains as abiotic systems. Many mountain systems have undergone multiple cycles of warm and cold climates in the past. The author could present several scenarios on how the population can overcome or adapt to changes in the mountains that are predicted or already occurring.
- The paper does not analyze the benefit of mountains as a source of mineral resources and energy for a growing humanity. How it is possible to preserve mountain systems while depleting resources such as ores, coal, stone and more.
- It should explain what soft infrastructure is all about.
- Even so, the author should be aware that not all mountain systems on Earth are endangered and that the problem of protecting mountains cannot be viewed solely as a global problem. As something that presents itself globally, perhaps a map of the world showing the most endangered "mountain regions" with the appropriate typology of the problem should be included in the paper.

Author Response

Thank you for reviewing my work. I found several of your suggestions helpful so I have accommodated them in the revised file. However there are aspects where we clearly differ regarding the most important role of mountains for humanity, i.e. mountains as an irreplaceable natural heritage worthy of ambitious conservation measures (my point), and mountains as resources for economic exploitation (your suggestion). I have offered specific responses below. In addition please check the highlighted parts of the manuscript to trace changes.

PS I have made substantial changes to the manuscript in response to all reviewer comments, kindly refer to sections highlighted in yellow to confirm changes. In addition I have inserted a note thanking all reviewers for their constructive feedback that made this work better.

Reviewer comment

Author response

- In the abstract it is necessary to show the goal (s) of the work.

Text has been inserted to clarify this point

- The paper presents interesting ideas about changes in the mountains that affect all the inhabitants of the Earth, so the topic of the paper is topical.
- The term "Mountain Regions" appears in the title but the text does not explain exactly what it means.

Explanation has been provided

- I don't see why reducing the ice cover would be a threat to the mountains as abiotic systems. Many mountain systems have undergone multiple cycles of warm and cold climates in the past. The author could present several scenarios on how the population can overcome or adapt to changes in the mountains that are predicted or already occurring.

It is precisely a problem because under anthropogenic global environmental change, the loss of ice/snow cover is NOT a part of the natural warm-cold cyclicity that you mentioned. Similarly, species have been lost in the past due to natural selection, but that is not the same with anthropogenic species extinction that is now occurring globally. The whole idea of global (anthropogenic) environmental change rests on the fact that it is NOT natural, and therefore these should not be confused.

I have clearly mentioned some effects of anthropogenic climate change related ice/snow loss on mountains such as negative impact on cold adapted biota, loss of freshwater service provisions over the long term etc. and there are a number of works such as Hoorn et al.’s Introductions in the ‘Mountains, Climate and biodiversity’ volume, and Immerzeel et al’s recent paper in Nature that can be cited (cited in my paper already). Snow/ice cover, glacial volume are all parts of the abiotic diversity of mountains, loss of those dimensions due to anthropogenic drivers both impoverishes abiotic diversity of mountains as well as jeopardizes species survival.

I have now also inserted a brief explanation to make things more obvious.

Finally, it is simply not a matter for human populations finding a clever way to adapt to changes, the dimension is bigger than involves the loss of natural heterogeneity and evolutionary pathways under cold adapted niches due to anthropogenic climate change.

- The paper does not analyze the benefit of mountains as a source of mineral resources and energy for a growing humanity. How it is possible to preserve mountain systems while depleting resources such as ores, coal, stone and more.

It is not the aim of my paper to uphold mountains as a possible resource that can be exploited for economic gains. The aim of the paper is to posit the importance of mountains as a global level natural heritage worthy of conservation.

In fact the very idea of natural heritage is to preserve those aspects of nature that are of great scientific value or intrinsically important for the biosphere. So as soon as anyone approaches mountains as a form of ‘natural heritage’ I believe the economic exploitation angle is necessarily precluded.

In addition, just let me remark that holding everything in the geo-biosphere as resources fit for exploitation under the onslaught of a ‘growing humanity’ is the core of the problem we face today in the Anthropocene, not its solution in any way, in my opinion.

- It should explain what soft infrastructure is all about.

A brief explanation has been provided

- Even so, the author should be aware that not all mountain systems on Earth are endangered and that the problem of protecting mountains cannot be viewed solely as a global problem. As something that presents itself globally, perhaps a map of the world showing the most endangered "mountain regions" with the appropriate typology of the problem should be included in the paper.

The point of this paper is to provoke a discussion under the hypothetical position that all mountains in the world (specifically the 12.5% of the terrestrial surface outside Antarctica termed as mountainous) as under severe and growing threat. The view is consistent to the literature sources examined. So, the view presented here is that all mountain regions are under some form of anthropogenic stress. Of course, this claim remains open to falsification, so anyone can falsify it by demonstrating that many mountains are not under any sort of threat under anthropogenic environmental change (which seems to be your suggestion)…but the main focus of this paper remains on arguing that mountains are a global level natural asset under incremental threat, so I do not see a reason to water down this claim.

Reviewer 2 Report

The author makes a call that mountains merit ambitious conservation schemes and I am very much supporting this call. However, from my point of view, the fact that the argument builds upon a somewhat eclectic and limited choice of evidence and information compromises the impact of the message. I am also doubtful about the impact of this paper, as it offers no suggestion on any policy framework or global initiative/convention to which scientists and others mentioned could speak. For publication and to really serve the mountain science community, this publication should in my opinion build more rigorously on existing literature and arguments and refer to other recent calls for placing mountains in global conservation and sustainable development agendas.

LINE 31-33: repetition of lines 28-30

LINE 33-36: unless policies are formulated and local populations and stakeholders involved, I am reluctant to believe that scientists, conservation professionals, tourism planners and communication experts can address the degradation of mountain environments.

LINE 75-77: this is not entirely correct as there is one specific SDG target 15.4 that pertains to safeguarding mountains

LINE 51-70: glacier loss is the traditional go-to story. As rightfully mentioned on lines 53-56, there are other drivers besides climate change in mountains, and impact of those do not converge on the loss of glacial ice volume and colonization by lowland species. I regret this myopic view of the effects of global change on mountains.

Section 3, first paragraph: while I agree that mountains might also be acknowledged as drivers of biodiversity, this section appears unnecessarily lengthy for driving this point home. For instance, I do not see the need for both citing Rahbeck and using the figure used in this paper in addition to the rest on mountain uplift. Additionally, some of this section pertains to mountains as storehouses of biodiversity, which is already the topic of section 2.  I would also argue that the first sentence of the section rather pertains to its second part on ecosystem services

Section 3, second paragraph: the mention of research and conservation focus is a bit surprising here as this has not been the lens used so far. The content of the paragraph comes across as a bit of cherry picking, with very few specific papers cited while the literature on the topic covered in this paragraph is vast and offers plenty of nuances.

Section 4, title: I don’t get from this section the notion of “shared” heritage. Why is it needed?

LINE 126-128: this sentence reads as if addressing the issue of connectivity of mountain, watershed, and lowland spaces would suffice to slow down degradation and decrease threat, which I don’t agree with.

LINE 128-132: there are many more initiatives that focus on mountains, their protection, sustainable development, etc. The selection made by the author is surprising and a note at least is needed to acknowledge the existence of many other initiatives.

The author makes suggestions on what could be done from a conservation point of view but what policy instrument would allow this at global scale and what global initiative could host a call for the solutions proposed by the author?

LINE 166-171: I don’t really see the pertinence or relevance of this example.

LINE 173-175: I think that the word “review” is not appropriate as the author merely brushes over a somewhat biased set of “threats”. I also don’t understand what the author means with “threat scenarios” as no scenario is discussed in this paper. I further argue that “the need to value mountain systems for their endemic diversity as well as numerous supporting roles” has been formulated in the past.

Author Response

Thank you for your comments to my paper. I partly agree with your criticism that the scope of the paper may appear narrow as there are so many aspects regarding mountains. However I wrote this as a ‘position paper’ in order to put forward a certain argument (i.e. of proactive conservation of mountains as a collective natural heritage)—which is a relatively under-addressed concept in current literature. It is framed as a short and selective review as per the journal requirement. Hence the references are also selective, and the paper highlights those works that help to uphold the core argument. And as the literature cited comes from reputed scientific journals, the argument is buttressed by relevant and reliable information. Of course, I remain fully aware that aspects such as culture, socioeconomic drivers, and local dynamics are important, but, as you pointed out, there are already many works available on those facets. The aim of this paper is to raise a (probably somewhat provocative) argument of proactive conservation of mountain systems as a natural heritage in the light of their ongoing degradation despite existing initiatives. To my knowledge this angle is relatively under-assessed in existing mountain literature and in this way the paper makes a contribution to the debate of how to better conserve mountains in today’s rapidly changing world. Of course, this perspective will require further problematization (or can be refuted by a counter-argument) but it is hoped that the call will help to emplace mountains as a conservation target for their intrinsic value.

Below, I have tried to respond to your specific comments.

PS I have made substantial changes to the manuscript in response to all reviewer comments, kindly refer to sections highlighted in yellow to confirm changes. In addition I have inserted a note thanking all reviewers for their constructive feedback that made this work better.

Reviewer comment

Author response

However, from my point of view, the fact that the argument builds upon a somewhat eclectic and limited choice of evidence and information compromises the impact of the message.

As noted above, it is designed as a ‘position paper’ that presents a selective literature survey in order to uphold an argument. I have clarified this in the revised version.

I am also doubtful about the impact of this paper, as it offers no suggestion on any policy framework or global initiative/convention to which scientists and others mentioned could speak.

For publication and to really serve the mountain science community, this publication should in my opinion build more rigorously on existing literature and arguments and refer to other recent calls for placing mountains in global conservation and sustainable development agendas. 

I have now mentioned the IUCN’s Mountains Specialist Group platform (following another reviewer’s suggestion) as well as the Connectivity Conservation Group as a starting point. The UNESCO World Natural Heritage Program, to which IUCN is an advisory body, remains probably the most useful existing protection platform for a natural heritage. I am a member of IUCN-WCPA myself, and I hope that IUCN can send out a much more proactive and ambitious call for preserving mountain ecosystems.

I have added some points related to the angles you raised.

LINE 31-33: repetition of lines 28-30

I have rephrased this section.

LINE 33-36: unless policies are formulated and local populations and stakeholders involved, I am reluctant to believe that scientists, conservation professionals, tourism planners and communication experts can address the degradation of mountain environments.

I agree that local populations are always important. My point is that despite strong focus on local involvement, nature conservation schemes have largely failed to deliver till date (a million species are threatened with extinction at this point, so change is on a negative trajectory), and probably a more proactive preservation focus is required. But I agree that even that will require local understanding so I have made a note of the need of local involvement.

LINE 75-77: this is not entirely correct as there is one specific SDG target 15.4 that pertains to safeguarding mountains

I added that information, but please note that Agenda 15.4 places mountains as subservient to sustainable development and the way SDGs are framed, it favors a growth/development oriented pathway, not conservation oriented outcomes. I have added that argument too.

LINE 51-70: glacier loss is the traditional go-to story. As rightfully mentioned on lines 53-56, there are other drivers besides climate change in mountains, and impact of those do not converge on the loss of glacial ice volume and colonization by lowland species. I regret this myopic view of the effects of global change on mountains.

Agreed there are many change pathways, but here the convergence angle is highlighted for the pressing need of understanding the threat scenario. This is not a myopic view, but a valid point that high mountain regions across the world are affected by this set of drivers.

Section 3, first paragraph: while I agree that mountains might also be acknowledged as drivers of biodiversity, this section appears unnecessarily lengthy for driving this point home. For instance, I do not see the need for both citing Rahbeck and using the figure used in this paper in addition to the rest on mountain uplift. Additionally, some of this section pertains to mountains as storehouses of biodiversity, which is already the topic of section 2.  I would also argue that the first sentence of the section rather pertains to its second part on ecosystem services

This is an important point so I believe it is worth the explanation. I have adjusted this portion somewhat to address you concern (the Rahbek citation has been moved upward) please check highlighted text.

Section 3, second paragraph: the mention of research and conservation focus is a bit surprising here as this has not been the lens used so far. The content of the paragraph comes across as a bit of cherry picking, with very few specific papers cited while the literature on the topic covered in this paragraph is vast and offers plenty of nuances.

Because it is a selective review I could not include all possible angles. Some information has been added in the revised version that will serve as further pointers.

Section 4, title: I don’t get from this section the notion of “shared” heritage. Why is it needed?

Because mountains collectively form a natural heritage which delivers benefits to all people in all continents. An explanatory note has been added. Besides, the importance of mountains as a global common good has been referred to, after the suggestion made by another reviewer.

LINE 126-128: this sentence reads as if addressing the issue of connectivity of mountain, watershed, and lowland spaces would suffice to slow down degradation and decrease threat, which I don’t agree with.

It will be a positive step nonetheless. Connectivity between mountains, watersheds, and lowland requires urgent attention, especially when formulating conservation plans for the integrity of mountains as a natural heritage. You may or may not agree with this viewpoint, but agreement/disagreement are personal predispositions.

LINE 128-132: there are many more initiatives that focus on mountains, their protection, sustainable development, etc. The selection made by the author is surprising and a note at least is needed to acknowledge the existence of many other initiatives.

New information has been added.

The author makes suggestions on what could be done from a conservation point of view but what policy instrument would allow this at global scale and what global initiative could host a call for the solutions proposed by the author?

While this remains something to be further explored, I have provided some more specific pointers.

LINE 166-171: I don’t really see the pertinence or relevance of this example.

There is obvious relevance! I have cited the well known case of camera and digital tracking of the snow leopard in the Himalayas, a keystone species in that mountain region.

LINE 173-175: I think that the word “review” is not appropriate as the author merely brushes over a somewhat biased set of “threats”. I also don’t understand what the author means with “threat scenarios” as no scenario is discussed in this paper. I further argue that “the need to value mountain systems for their endemic diversity as well as numerous supporting roles” has been formulated in the past.

As mentioned several times before, this concise review formulates an argument of mountains as a global level natural heritage. I am not sure this has been explicitly argued in publications in the past, even if so, such publications are rare, and therefore this work has its value.

]

Reviewer 3 Report

Overall, sections 1 to 3 of this paper are an excellent synthesis, mainly based on recent literature.  However, the main idea presented in the paper is not entirely new: e.g., it should refer to previous literature about mountains as a global common good (e.g., https://www.tandfonline.com/doi/full/10.1080/14650040701783375 , https://www.press.uchicago.edu/ucp/books/book/chicago/M/bo15302284.html ).  With regard to ecosystem services, a recent review ( https://journals.plos.org/plosone/article?id=10.1371/journal.pone.0217847 ) should be cited.  Citation 4 has been overtaken by citation 17; and references to books (e.g., 31) should, where possible, refer to specific chapters.

Section 4 makes some good arguments, but needs revision.  There is a certain inconsistency between the title of the paper and the name of this section.  If the name of this section remains as it is, I would like to see a better balance between natural heritage (the current emphasis) and cultural heritage and, in any case, recognition that a large proportion of mountain landscapes are cultural landscapes, and that the activities of people - past and present - have been and remain key driving forces of biological diversity.  I am also surprised that it includes nothing about the very long-standing mountains group of the IUCN World Commission on Protected Areas, which also includes a specialist group on connectivity conservation.  The GMBA focuses on biodiversity; but the IUCN mountain group focuses on protected areas.  This needs to be remedied when the paper is revised.  Another point that should be made is that beyond the designation protected areas and networks, there is an essential need for effective governance (e.g., see https://www.sciencedirect.com/science/article/pii/S1462901114001920 and other articles in this special issue and https://agupubs.onlinelibrary.wiley.com/doi/abs/10.1029/2018EF001024 ).  Furthermore, World Heritage Sites are only one of a number of global approaches to maintain mountain heritage; there are others, mainly under the umbrella of UNESCO; but also the Ramsar Convention. 

Author Response

Thank you very much for reviewing my paper. I found your feedback very constructive and helpful for improving my work. I also found the resources you suggested as very pertinent and have incorporated them into the revised version. I have accommodated nearly all the suggestions you had for the paper, but kindly note that the focus of this short piece remains on the natural aspect of mountain environment so the cultural dimension could not be discussed. I have made this explicit in the revised version, and have appended the word ‘natural’ to heritage in order to avoid confusion and to better justify the approach.

Please see my responses to your specific comments below.

PS I have made substantial changes to the manuscript in response to all reviewer comments, kindly refer to sections highlighted in yellow to confirm changes. In addition I have inserted a note thanking all reviewers for their constructive feedback that made this work better.

Reviewer comment

Author response

However, the main idea presented in the paper is not entirely new: e.g., it should refer to previous literature about mountains as a global common good.  With regard to ecosystem services, a recent review  should be cited. 

Thank you for suggesting these sources. They have now been included into the revised version.

Citation 4 has been overtaken by citation 17; and references to books (e.g., 31) should, where possible, refer to specific chapters.

Modified in the revised version.

Section 4 makes some good arguments, but needs revision.  There is a certain inconsistency between the title of the paper and the name of this section.  If the name of this section remains as it is, I would like to see a better balance between natural heritage (the current emphasis) and cultural heritage and, in any case, recognition that a large proportion of mountain landscapes are cultural landscapes, and that the activities of people - past and present - have been and remain key driving forces of biological diversity. 

I have revised section 4. While I agree that a large part of mountain landscapes is cultural, this short review is aimed at arguing for the natural heritage angle in the light of rapid change of the natural environment of mountains. I have added sentences to make this explicit and to avoid confusion.

I am also surprised that it includes nothing about the very long-standing mountains group of the IUCN World Commission on Protected Areas, which also includes a specialist group on connectivity conservation.  The GMBA focuses on biodiversity; but the IUCN mountain group focuses on protected areas.  This needs to be remedied when the paper is revised.  Another point that should be made is that beyond the designation protected areas and networks, there is an essential need for effective governance and other articles in this special issue and Furthermore, World Heritage Sites are only one of a number of global approaches to maintain mountain heritage; there are others, mainly under the umbrella of UNESCO; but also the Ramsar Convention.

I have now mentioned the IUCN Mountains and Connectivity Conservation groups as well as the MAB and Ramsar conventions briefly. I am a member of the WCPA myself and I do hope that IUCN bodies can deliver better conservation outcomes for mountains.

Round 2

Reviewer 2 Report

I appreciate the effort invested in addressing my comments.  

I still don't see the need for Figure 1 and Figure 2. References to those publications suffice.

The notion of "threat scenario" remains to be clarified. 

Author Response

Thank you for your comments. I have addressed them below: 

  1. Re the figures: I believe they are important for readers without specific expertise in this field to understand the salient features of the linkage between abiotic-biotic diversity in mountain environs (Fig 1) and the extent of mountains as a global heritage (Fig 2). Hence I have chosen to retain them, unless there is any copyright related issue with reproduction.
  2. Re the 'threat scenario': This was actually elaborated on in Section 2: Mountains as storehouses of natural diversity under threat', particularly in lines 73-97. I have further spelled out the main contours of the threat scenario in the conclusion to avoid any confusion (please see added text in lines 267-270). 

Some wording and punctuation errors were corrected in the revised form.